# Structure and function of the metagenomic plastic-degrading polyester hydrolase PHL7 bound to its product

P. Konstantin Richter[1], Paula Blázquez-Sánchez[2], Ziyue Zhao[2], Felipe Engelberger [3], Christian Wiebeler [2,4], Georg Künze [3], Ronny Frank [5], Dana Krinke[5], Emanuele Frezzotti[6], Yuliia Lihanova[2], Patricia Falkenstein [2], Jörg Matysik [2], Wolfgang Zimmermann [2] ✉, Norbert Sträter [1] ✉ & Christian Sonnendecker [2] ✉

The recently discovered metagenomic-derived polyester hydrolase PHL7 is able to efficiently degrade amorphous polyethylene terephthalate (PET) in post-consumer plastic waste. We present the cocrystal structure of this hydrolase with its hydrolysis product terephthalic acid and elucidate the influence of 17 single mutations on the PET-hydrolytic activity and thermal stability of PHL7. The substrate-binding mode of terephthalic acid is similar to that of the thermophilic polyester hydrolase LCC and deviates from the mesophilic *Is*PETase. The subsite I modifications L93F and Q95Y, derived from LCC, increased the thermal stability, while exchange of H185S, derived from *Is*PETase, reduced the stability of PHL7. The subsite II residue H130 is suggested to represent an adaptation for high thermal stability, whereas L210 emerged as the main contributor to the observed high PET-hydrolytic activity. Variant L210T showed significantly higher activity, achieving a degradation rate of 20 $\mu$m h$^{-1}$ with amorphous PET films.

The global plastic production gained traction by the end of World War II and skyrocketed since then. By 2015, global polyethylene terephthalate (PET) resin production reached 33 million tons, accounting for 8.7% of the annual plastic production of 381 million tons[1]. A total of 79% of plastic waste produced since 1950 ended up in landfills or natural environments such as the ocean, while only 9% have been recycled[1]. The longevity of plastic products and the careless handling of the respective waste has caused significant world-wide environmental and health problems[2]. Therefore, there is an urgent need to develop sustainable and cost-effective plastic waste recycling technologies. Biocatalytic plastic recycling may become an interesting environmentally friendly and energy-efficient technology to cope with the global plastic waste problem[3].

PHL7 is a thermophilic polyester hydrolase isolated from a plant compost metagenome along with six homologs (PHL1-6)[4]. It rapidly hydrolyzes amorphous PET at 70 °C yielding terephthalic acid (TPA) and ethylene glycol (EG), outperforming all previously reported PET-hydrolytic enzymes, including engineered variants[5,6]. Initially generated mono-(2-hydroxyethyl) terephthalic acid (MHET) is further hydrolyzed to TPA and EG by PHL7, while no significant amounts of bis-(2-hydroxyethyl) terephthalic acid (BHET) are released during PET hydrolysis[4]. PHL7 is therefore a candidate for the development of biocatalytic recycling processes of PET waste.

[1]Institute of Bioanalytical Chemistry, Centre for Biotechnology and Biomedicine, Leipzig University, Leipzig, Germany. [2]Institute of Analytical Chemistry, Leipzig University, Leipzig, Germany. [3]Institute for Drug Discovery, Leipzig University Medical School, Leipzig, Germany. [4]Wilhelm-Ostwald-Institute for Physical and Theoretical Chemistry, Leipzig University, Leipzig, Germany. [5]Centre for Biotechnology and Biomedicine, Molecular Biological-Biochemical Processing Technology, Leipzig University, Leipzig, Germany. [6]Department of Chemical Life and Environmental Sciences, University of Parma, Parma, Italy. ✉e-mail: wolfgang.zimmermann@uni-leipzig.de; strater@bbz.uni-leipzig.de; christian.sonnendecker@uni-leipzig.de

PHL7 enqueues in a number of microbial polyester hydrolases with similar properties. *HiC* cutinase[7,8], *Tf*Cut2[9,10], Cut190[11–13], Est119[14–17] and LCC[5,18,19] hydrolyze their substrates at temperatures above 50 °C, with optimal reaction temperatures of up to 72 °C in the case of LCC. Other polyester hydrolases such as *Is*PETase[20–28], PE-H[29] or Mors1[30] are produced by mesophilic or psychrophilic bacteria with a temperature optimum between 25 and 40 °C.

PHL7 in turn is stable in a temperature range suitable for PET hydrolysis of 65–70 °C and hydrolyzes PET most effectively around 70 °C, which is the glass transition temperature $T_g$ of the polymer[4]. Catalytic activity at an elevated temperature is a prerequisite for an efficient enzymatic hydrolysis of PET[3,31,32]. However, one significant drawback remains: PET is composed of amorphous and crystalline domains depending on its manufacturing process, but none of the polyester hydrolases characterized to date is able to degrade crystalline PET found e.g., in beverage bottles and textile fibers in any significant amounts[4,7,20,32–34]. Biocatalytic degradation of crystalline PET therefore requires an energy-intensive pre-treatment to amorphize the material[5,32].

All polyester hydrolases share an α/β-hydrolase structure with a canonical catalytic triad comprised of serine, histidine and aspartate[4,10,16,19,21,35,36]. Despite their structural conservation, they differ significantly in their PET-hydrolytic activity and thermal stability.

Polyester hydrolases were categorized as type I, IIa and IIb based on differences in subsite II of the catalytic binding pocket[22] (Fig. 1). Enzymes of different types vary especially in the two positions equivalent to H130 (the residue prior to the catalytic serine) and L210 in PHL7. A further difference is the presence or absence of an extended loop connecting β-strand 8 with the neighboring α-helix 5 and a second disulfide bridge connecting this loop with the adjacent β-strand 7. Type I polyester hydrolases such as LCC contain a histidine prior to the catalytic serine, a phenylalanine in subsite II and lack the extended loop and the second disulfide bridge. Type II enzymes contain the aforementioned extended loop in subsite II and a second disulfide bond between this loop and β−strand 7 (C$_{203}$-C$_{239}$ in *Is*PETase, Fig. 1b).

They also display tryptophan instead of histidine prior to the catalytic serine and either F/Y (type IIa) or S (type IIb) in the position equivalent to L210 in PHL7. Both the extended loop and the disulfide bridge occur together in the mesophilic polyester hydrolase *Is*PETase. Elongation of that particular loop increases the flexibility of this region and, as a consequence, compromises the integrity of the catalytic triad. The additional disulfide bridge C$_{203}$-C$_{239}$ then restricts the higher flexibility of the enzyme in the active site caused by the elongated loop. Reduction or mutagenesis of this disulfide bridge decreases the PET-hydrolytic activity of *Is*PETase[23].

The aim of our study is to investigate the binding mode of TPA to PHL7 and to compare the cocrystal structure with mesophilic (*Is*PETase) and thermophilic (LCC) homologs. Single key residues of PHL7 were substituted to mirror corresponding amino acids of the LCC or *Is*PETase substrate binding sites (Supplementary Table 1 and Supplementary Fig. 1), and their effects on thermal stability and PET-hydrolytic activity were examined. We further substituted L210 of PHL7, a key residue for PET-hydrolytic activity, with a set of amino acids and investigated the binding contribution of individual residues by molecular docking and quantum-mechanical calculations.

## Results and discussion

### TPA binds to subsite I of the active site of PHL7

We cocrystallized the inactive variant PHL7 S131A along with the PET hydrolysis product TPA (Figs. 1 and 2, Supplementary Table 2). The molecule binds to the active site close to S/A131. Its proximal carboxyl group interacts with the oxyanion hole (formed by the two backbone nitrogen atoms of F63 and M132), the ε-nitrogen of H209 and two water molecules. These interactions, which are optimized to stabilize the tetrahedral sp$^3$-hybridized transition state structure, rotate the proximal carboxylate group by 40° out of the plane of the aromatic ring (Fig. 2b). Such a deviation from the in-plane lowest energy conformation has also been observed for complex structures of *Is*PETase or LCC (Fig. 2c, d). The aromatic phenylene ring of TPA is embedded in

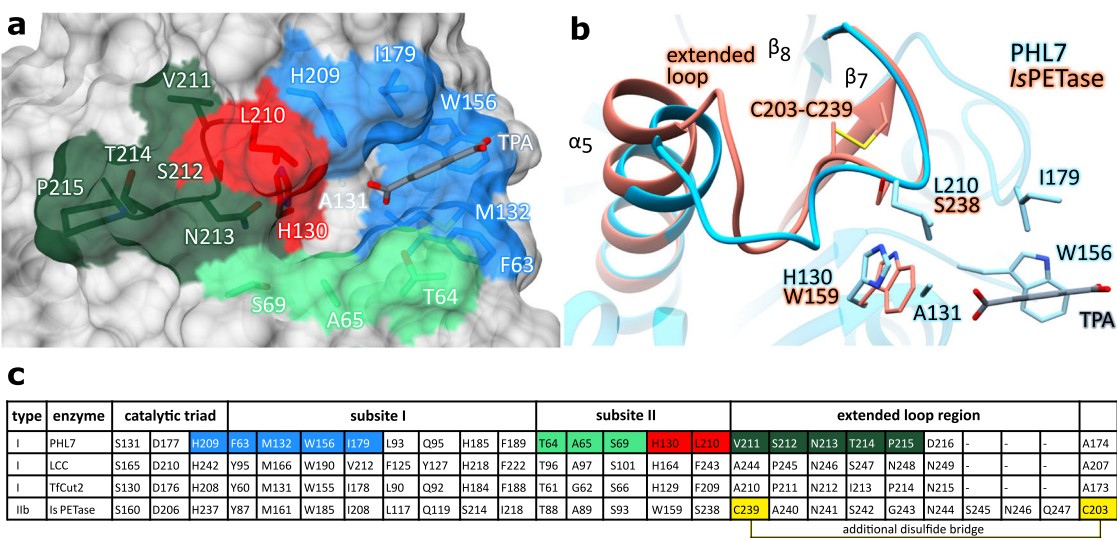

| type | enzyme | catalytic triad | | | subsite I | | | | | | | | subsite II | | | | | extended loop region | | | | | | | | |
|---|---|---|---|---|---|---|---|---|---|---|---|---|---|---|---|---|---|---|---|---|---|---|---|---|---|---|
| I | PHL7 | S131 | D177 | H209 | F63 | M132 | W156 | I179 | L93 | Q95 | H185 | F189 | T64 | A65 | S69 | H130 | L210 | V211 | S212 | N213 | T214 | P215 | D216 | - | - | - | A174 |
| I | LCC | S165 | D210 | H242 | Y95 | M166 | W190 | V212 | F125 | Y127 | H218 | F222 | T96 | A97 | S101 | H164 | F243 | A244 | P245 | N246 | S247 | N248 | N249 | - | - | - | A207 |
| I | TfCut2 | S130 | D176 | H208 | Y60 | M131 | W155 | I178 | L90 | Q92 | H184 | F188 | T61 | G62 | S66 | H129 | F209 | A210 | P211 | N212 | I213 | P214 | N215 | - | - | - | A173 |
| IIb | Is PETase | S160 | D206 | H237 | Y87 | M161 | W185 | I208 | L117 | Q119 | S214 | I218 | T88 | A89 | S93 | W159 | S238 | C239 | A240 | N241 | S242 | G243 | N244 | S245 | N246 | Q247 | C203 |

additional disulfide bridge

**Fig. 1 | Subsites I and II of polyester hydrolases. a** Surface representation of PHL7×TPA. TPA (gray) binds to subsite I (blue). Residues comprising the putative subsite II are conserved or semi-conserved (light green), cover a loop that is elongated in mesophilic homologs (dark green) or differ significantly between polyester hydrolases (red). **b** Superimposition of PHL7×TPA (light blue) with *Is*PETase (salmon, PDB ID: 5XJH). An extended loop covering residues C239-Q247 of *Is*PETase deviates from its equivalent region (V211-D216) of PHL7 and is stabilized by a non-conserved disulfide bridge (C203-C239). Activity-regulating L210 lies upstream of that loop and spatially close to H130. **c** Comparison of structurally equivalent active site residues of PHL7 and its homologs LCC, *Tf*Cut2 and *Is*PETase. The catalytic triad and subsite I residues are conserved or conservatively substituted. Residues equivalent to L93, Q95, H185 and F189 in PHL7 can be considered as a part of subsite I although they do not directly interact with TPA in the cocrystal structure. Subsite II residues and the adjacent loop deviate more significantly, especially among thermophilic (type I) and mesophilic (type IIb) homologs. The loop-stabilizing disulfide bridge of *Is*PETase is not conserved in type I polyester hydrolases.

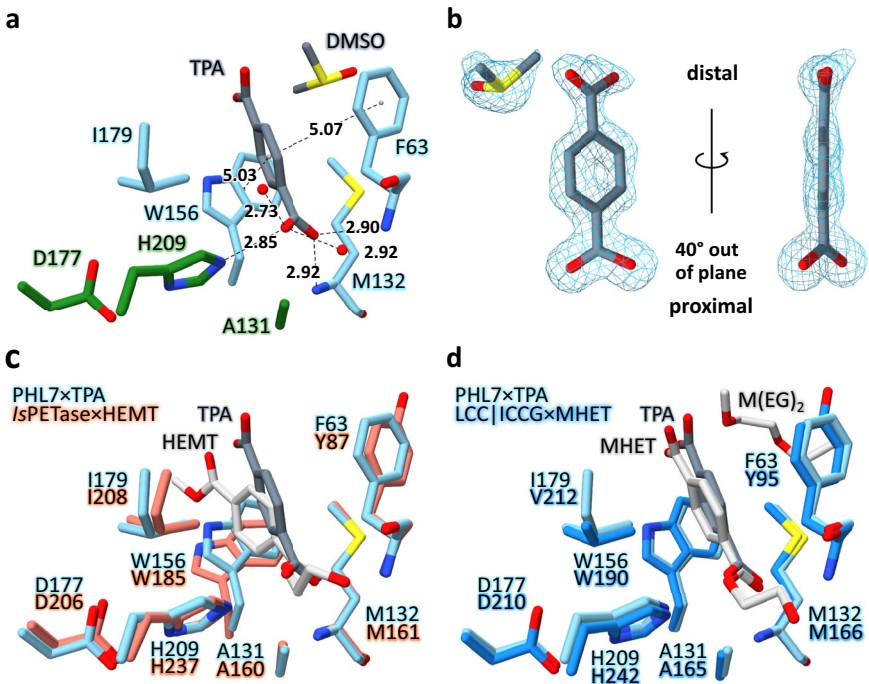

**Fig. 2 | Subsite I of PHL7×TPA. a** TPA (dark gray) interacts with F63, W156 (π-stacking), I179, the backbone nitrogen atoms of F63 and M132 and $N_\epsilon$ of H209. Residues A131 (S131 in PHL7 WT), D177 and H209 form the catalytic triad (green, other residues in light blue). All distances in Å. **b** (2$F_o$-$F_c$)-type electron density of TPA and DMSO, contoured at 1.5 σ. The proximal carboxyl group of TPA is rotated out of the aromatic plane by 40°. **c** Superimposition of PHL7×TPA subsite I (light blue) and *Is*PETase×HEMT (salmon, PDB ID: 5XH3). Most structurally equivalent residues align well except for W156 (W185 in *Is*PETase). The aromatic ring of HEMT (light gray) bound to *Is*PETase is displaced and rotated relative to the respective ring of TPA (dark gray) bound to PHL7. **d** Superimposition of PHL7×TPA subsite I (light blue) and LCC | ICCG×MHET (blue, PDB ID: 7VVE). All equivalent residues are well conserved. MHET (light gray) bound to LCC | ICCG aligns well on TPA (dark gray) bound to PHL7. M(EG)$_2$ (diethylene glycol monomethyl ether, light gray) binds close to subsite I of LCC. DMSO was omitted in **c** and **d**.

## Table 1 | Overview of ligand-cocrystal structures of polyester hydrolases

| Enzyme | Organism | Opt. reaction temperature[a] | Uniprot ID | PDB ID | Ligand |
|---|---|---|---|---|---|
| PHL7 | metagenome | 70 °C[4] | — | 8BRB (this study) | TPA |
| LCC | metagenome | 50–72 °C[5,18] | G9BY57 | 6THS[5] 6THT[5] 7DS7[b] 7VVE[38] | 1,4-diethylene oxide imidazole imidazole MHET |
| *Tf*Cut2 | *Thermobifida fusca (KW3)* | 50–60 °C[9,76] | E5BBQ3 | 4CG2[10] | phenylmethanesulfonic acid |
| Est119 | *Thermobifida alba (AHK119)* | 50 °C[14] | F7IX06 | 6AID[17] | lactate, ethyl-D-lactate |
| Cut190 | *Saccharomonospora viridis* | ~50 °C (WT)[11] ~70 °C (S226P) | W0TJ64 | 5ZRR[12] 5ZRS[12] 7CTR[13] 7CTS[13] | monoethyl succinate monoethyl adipate 1,4-diethylene oxide 1,4-diethylene oxide |
| *Hi*C | *Humicola insolens* | 70 °C[7] | | 4OYL[8] | monoethyl phosphate |
| PETase | *Ideonella sakaiensis* | 30–40 °C[20] | A0A0K8P6T7 | 5XH2[21] 5XH3[21] | p-nitrophenol HEMT |

[a]The optimal reaction temperature of polyester hydrolases depends on factors such as salt concentration, pH value or stabilizing mutations. Therefore, mostly temperature ranges are depicted.
[b]No primary literature currently published for this structure. https://doi.org/10.2210/pdb7DS7/pdb.

a hydrophobic binding pocket comprised of the side chains of F63, M132, W156, and I179 (subsite I[22], Fig. 2a).

The benzene moiety adopts a distorted T-shaped π-π-interaction with W156 at a distance of 5.03 Å and an angle of 63.2° and with F63 at a distance of 5.07 Å and an angle of 65.7°, respectively. M132 and I179 complete the hydrophobic environment in subsite I. The distal carboxyl group does not interact with the protein but only with adjacent water molecules and DMSO that derives from the crystallization buffer. The two protein chains in the asymmetric unit are highly similar (Supplementary Fig. 2). Cocrystal structures of homologous polyester hydrolases

containing MHET, 1-(2-hydroxyethyl) 4-methyl terephthalate (HEMT) or ligands that are not building blocks of PET have also been described (Table 1).

The overall folds of PHL7 and PHL7×TPA are nearly identical, indicated by an all-atom $C_\alpha$ RMSD of 0.28 ± 0.09 Å (mean value calculated by comparison of all chains that are present in the asymmetric units of both crystal structures ± standard deviation). Additionally, the orientations of active site amino acids of PHL7×TPA does not deviate significantly from apo PHL7 (PDB ID: 7NEI, Supplementary Fig. 3). TPA binding therefore does not induce a significant conformational change.

Substrate-bound PHL7×TPA and *Is*PETase×HEMT (PDB ID: 5XH3[21], Fig. 2c) deviate more significantly from each other, indicated by an increased all-atom $C_\alpha$ RMSD of 1.36 ± 0.00 Å. Subsite I of the substrate-binding groove, to which both TPA and HEMT bind, is wider open in *Is*PETase×HEMT than in PHL7×TPA. The HEMT molecule bound to *Is*PETase pushes W185 (W156 in PHL7) away from the active site and into conformation B[21], which is possibly due to the small serine side chain in the adjacent position 214. Besides the conformational change of loop β7–α5 upon substrate binding in *Is*PETase, MD-simulations from da Costa et al. further identified loop β1–β2 to be very flexible[37]. The ligand-induced opening of the substrate-binding groove is not possible in PHL7 since the flexibility of subsite I is decreased by H185 (S214 in *Is*PETase, Supplementary Fig. 4b), which hinders W156 to move away from the active site. This widening of subsite I in *Is*PETase allows for a shift of HEMT away from the protein surface and deeper into the hydrophobic binding pocket in comparison to TPA in PHL7. The two ligands deviate by 1.06 Å (proximal carboxylic carbon) to 2.96 Å (distal carboxylic carbon).

PHL7×TPA and the engineered LCC variant ICCG×MHET (PDB ID: 7VVE[38], Fig. 2d) superimpose at an all-atom $C_\alpha$ RMSD of 1.14 ± 0.03 Å. Structurally equivalent residues of subsite I as well as the ligands align without major deviations. One molecule of diethylene glycol mono-methyl ether [M(EG)$_2$] binds close to subsite I of LCC, adjacent to Y95 and Y127G (F63 and Q95 in PHL7, respectively). This site is occupied by DMSO in the PHL7 cocrystal structure. TPA and MHET deviate by 0.28 Å (proximal carboxylic carbon) to 0.95 Å (distal carboxylic carbon).

## Metal-binding sites

We previously described a bound metal ion in the crystal structure of unliganded PHL7 coordinated by E148, D233, the carbonyl group of F230 and three water molecules[4]. Due to the presence of 100 mM Na$^+$ in the crystallization buffer, the ion was refined as a sodium ion in the crystal structure. Addition of Ca$^{2+}$ or Mg$^{2+}$ improves the thermostability of PHL7 by up to 7 °C[4]. This phenomenon was also observed for other polyester hydrolases from thermophilic microorganisms[12,15,19,35,39]. The influence of Ca$^{2+}$ or Mg$^{2+}$ on the thermostability of mesophilic *Is*PETase has not been investigated. Metal binding sites in polyester hydrolases have been characterized for Cut190[12], *Tf*Cut2[39], Est119[15,40] and LCC[41].

The influence of metal ions on the stability of polyester hydrolases may in part result from the general stabilization of proteins by salts according to the Hofmeister series. However, individual metal binding sites may also provide additional specific stabilization. The coordination of the above mentioned metal ion binding site in PHL7 comprises an electrostatic interaction with the two negatively charged side chains of E148 and D233. It is widely accepted that, among other factors, favorable electrostatic interactions between positively (Arg, Lys) and negatively charged (Glu, Asp) residues can increase the thermal stability of proteins[42,43]. The thermophilic polyester hydrolase *Tf*Cut2 was previously stabilized by such substitutions, resulting in an increased melting temperature of up to 15.0 °C[39]. Therefore, we exchanged D233 by lysine in order to replace the sodium-mediated interaction by an electrostatic interaction between K233 and E148. The $T_m$ of PHL7 increased by 0.9 °C upon substitution of D233 with lysine, from 79.1 to 80.0 °C, and the PET-hydrolytic activity also increased slightly with respect to the WT. Further reports also describe an increased thermostability by introducing a disulfide bridge in *Tf*Cut2[44], Cut190[13] (site 2) or LCC[5] at positions of metal binding sites.

In a crystal structure of PHL7 that was obtained in the presence of 10 mM Mg$^{2+}$ ions, we identified a magnesium-binding site comprised of E13, D246 and D247. The Mg$^{2+}$ ion is complexed by the carboxyl side-chain of D246 and five water molecules in the first interaction shell forming an octahedral geometry. Three of the five water molecules in turn interact with the sidechains of E13, D246, and D247 while the other two are only coordinated by the divalent metal ion

(Supplementary Fig. 5). This Mg$^{2+}$-binding site of PHL7 resembles a Zn$^{2+}$-binding site identified in the thermophilic homolog Cut190 (binding site 4), which was described as a site unrelated to PET-hydrolytic activity[12]. In summary, two metal binding sites have been identified at the surface of PHL7. Similar to previous studies on other polyester hydrolases, these may as well serve as future mutagenesis targets to increase the thermal stability of PHL7. The two metal binding sites are 26 Å apart and may act independently and thus in an additive manner to increase protein stability.

## Functional analysis and systematic mutagenesis of subsite I residues

Subsite I of PHL7 is composed of five amino acids that directly interact with the substrate (F63, M132, W156, I179 and H209, Fig. 1a, c), and four residues that can be classified as a second interaction sphere influencing the conformation and flexibility of the aforementioned binding residues (L93, Q95, H185, F189). We compared the amino acid composition of subsite I of PHL7 with the homologous polyester hydrolases *Is*PETase, LCC and *Tf*Cut2 (both WT and engineered variants) and mutated selected residues to match them (Supplementary Fig. 1).

Disruption of the aromatic π-stacking clamp comprised of F63 and W156 decreased the PET-hydrolytic activity of PHL7 drastically as demonstrated by the F63A mutation. We exchanged F63 by both A and Y. The catalytic activity of the F63A variant dropped to 16% in comparison to the wild type whereas only a slightly reduced activity was observed for F63Y (after 4 h, Fig. 3a). Substitution at this position with alanine also led to reduced activity in *Is*PETase and LCC[5,22]. This finding underlines the importance of the aromatic residues Y87 and W185 of *Is*PETase (equivalent to F63 and W156 in PHL7) for the binding and hydrolysis of BHET. In the apo *Is*PETase structure (PDB ID: 5XG0), residue W185 is oriented in three different conformations (three individual protein chains) which has been interpreted as wobbling in this position. The equivalent residue W156 in PHL7 WT resembles conformation C of *Is*PETase W185 (Supplementary Fig. 4b). These multiple conformations in *Is*PETase are possible due to the neighboring residue S214. The small serine side chain in *Is*PETase allows W185 to move freely and adopt multiple conformations. Constricting this freedom of movement by mutagenesis in the S214H variant decreased the release of MHET by 50% and of TPA by 90%[21]. In PHL7, this histidine residue is present in the WT enzyme (H185, Fig. 1c). Mutating this position in PHL7 (H185S) in order to investigate the effect of increasing the mobility of the neighboring W156 destabilized PHL7 significantly, indicated by a $T_m$ decrease of 11.5 °C from 79.1 °C (WT) to 67.6 °C (H185S) (Fig. 3d). This mutation also decreased the PET-hydrolytic activity of PHL7 drastically to a remaining 22 and 15% after 4 and 8 h of reaction, respectively (Fig. 3a). This result was expected since the reaction temperature of 70 °C was above the melting temperature of the mutant. However, also at lower temperatures (50 °C and 60 °C) and a reaction time of 16 h, PHL7 WT outperformed variant H185S by a factor of 2.5 and 3.2, respectively (Fig. 3b).

Chen et al. analyzed the immediate vicinity of the amino acid residue S214 of *Is*PETase and found an isoleucine at position 218 deviating from a conserved phenylalanine in the structurally equivalent position of the thermophilic polyester hydrolases (F189 in PHL7)[45]. They proposed the amino acid pair S214/I218 to be crucial for PET-hydrolytic activity at ambient temperatures. In fact, the activity dropped significantly in the *Is*PETase S214H/I218F double mutant. An introduction of this H184S/F188I double mutant into the thermophilic homolog *Tf*Cut2 greatly enhanced its PET-hydrolytic activity at 60 °C[45]. This effect was also observed for the respective LCC H218S/F222I double mutant at reaction temperatures up to 60 °C. In contrast, the activity of the LCC double mutant decreased at temperatures above 60 °C[45]. This suggests a destabilizing effect of the S/I double mutant explaining the lower activity observed at elevated temperatures. An introduction of the single mutation F189I in PHL7 decreased its $T_m$ by

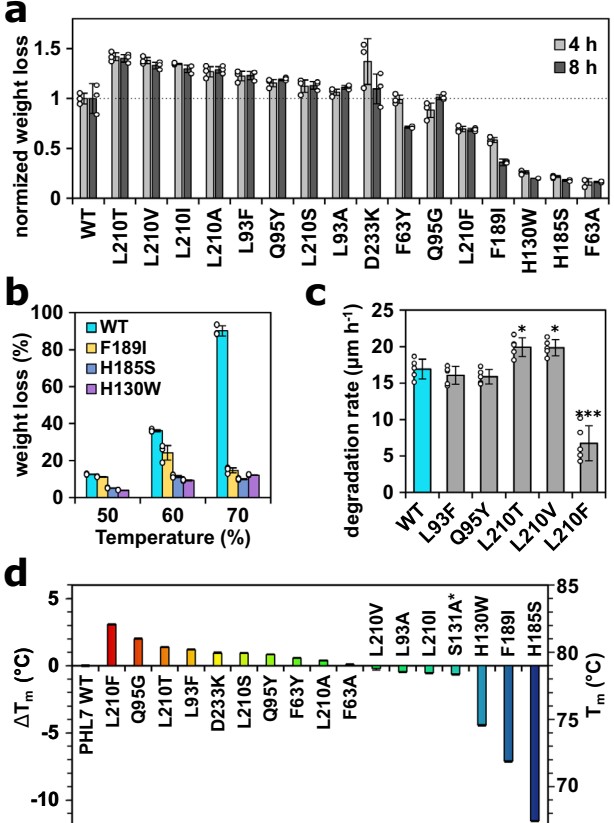

**Fig. 3 | PET film degradation and thermal stability of PHL7 variants. a** PET film weight loss after reaction times of 4 h and 8 h at 70 °C, data normalized to PHL7 WT. **b** PET film weight loss (in weight %, after 16 h) of PHL7 WT compared to variants with reduced thermal stability at lower reaction temperatures. **c** Analysis of PET film thickness reduction between 10 h to 15 h of reaction by impedance spectroscopy (*n* = 5 replicates). **d** Melting temperatures of PHL7 WT and its variants. The primary y-axis shows the difference in $T_m$ between the WT and variants, the secondary y-axis depicts the absolute melting temperature. If not other stated, mean values for *n* = 3 replicates ± SD are shown. * PHL7 variant S131A did not cause any PET weight loss after 24 h.

7.1 °C (from 79.1 to 72.0 °C) (Fig. 3d) and its PET-hydrolytic activity to 58 and 17% at 70 °C and reaction times of 4 h and 8 h, respectively. The activity of PHL7 F189I determined at 50 °C equaled the WT enzyme while it decreased by about 25% at 60 °C (Fig. 3b).

Mutational studies of three out of the four core subsite I residues in LCC (Y95, W190 and V212 single mutants, corresponding to F63, W156 and I179 in PHL7) resulted in a remaining activity of less than 48% in comparison to LCC WT, supporting our findings for PHL7[5]. Exchange of M161 or I208 (M132 and I179 in PHL7) to alanine reduced the BHET-hydrolytic activity in *Is*PETase to 52% and 46%, respectively[22]. This indicates that the π-π-interaction contributes more to a productive binding of BHET to subsite I than the two spatially adjacent aliphatic residues. The methionine residue (corresponding to M132 in PHL7, following the catalytic serine in position 131) is conserved among most polyester hydrolases. In the M132W variant, a substitution also found in PHL4[4], PET-hydrolytic activity is lost almost completely with a 97% reduction after a reaction time of 24 h (Supplementary Table 1). The bulky side chain of tryptophan likely occupies the space between F63 and W156. We assume that the aromatic clamp interacts with W132 in the M132W variant instead of π-stacking the first TPA moiety in subsite I. Therefore, productive binding of a PET chain was hindered and hydrolysis could not take place.

The amino acid residues L93 and Q95 are located in close proximity to the hydrophobic binding pocket of subsite I and vary between

PHL7 and LCC (Fig. 1c) but are conserved among most other polyester hydrolases[4]. Tournier et al. described F125 and Y127 (corresponding to L93 and Q95 in PHL7) as part of the first contact shell in substrate docking experiments with LCC[5]. LCC variant Y127G showed an increased thermostability by 2.3 °C, while the residual activity dropped to 67%[6]. An exchange of the corresponding PHL7 residues to aromatic amino acids (L93F and Q95Y) slightly improved the thermal stability (Fig. 3d), however a significant increase in activity could not be observed by electrochemical impedance spectroscopy (Fig. 3c). The exchange of L117F or Q119Y in *Is*PETase (L93 or Q95 in PHL7) increased the thermostability by 3 °C or 4.5 °C, respectively[28]. In contrast to our findings, the mutations L117F and Q119Y in *Is*PETase decreased its PET-hydrolytic activity by 30% or 50%, respectively[28]. This disparity could be due to the different optimal reaction temperatures which were 37 °C for *Is*PETase and 70 °C for PHL7[4,28].

The amino acid composition of the first interaction shell in subsite I of polyester hydrolases is therefore conserved with only minor differences among the thermophilic enzymes (PHL7 and LCC) and the mesophilic *Is*PETase. Disruption of the π-clamp decreased the activity drastically, while exchange of other amino acids impacts PET hydrolysis and stability of the protein to minor extents. The two residues H185 and F189 adjacent to W156 are a prerequisite for the thermal stability of PHL7 above 60 °C (Fig. 3d). Concurrently, this is also the case in the respective double mutants of the thermophilic polyester hydrolases *Tf*Cut2 and LCC[45].

## Variations in subsite II influence both activity and stability of PHL7

In contrast to subsite I, the architecture and size of subsite II differs significantly between polyester hydrolases, and the mode of binding of the PET chain has been debated controversially[22,46,47], as outlined below. Presently, no cocrystal structure of a PET oligomer binding to subsite II is available, most likely due to the hydrophobicity and low solubility of molecules comprised of more than one TPA moiety. Therefore, information on substrate binding to subsite II is mostly based on computational methods. Potential binding of three to five moieties has been described for Cut190, while mutational studies along the binding cleft showed most drastic effects for subsite I residues[46]. Subsite II of the thermophilic homolog LCC has been suggested to interact with two TPA moieties[5], while up to three PET building blocks were modeled into the respective region of the mesophilic enzyme *Is*PETase[22,24]. The latter binding mode has been challenged in a subsequent study investigating the conformation of PET chains at 30 °C and concluding that the OC-CO dihedral angle of the EG moiety detected via NMR spectroscopy is not in agreement with the conformation of the modeled PET 4-mer[47]. A recent study provided evidence that *Is*PETase WT prefers to bind PET in the *gauche* conformation, while the S238A variant is also able to bind PET in the *trans* conformation[48]. However, it is still not clear how the PET polymer chain binds to subsite II of polyester hydrolases, how many moieties interact with the protein and if binding differs between mesophilic and thermophilic homologs.

We created single mutants to investigate the influence of the aforementioned residues H130 and L210 on both protein stability and PET-hydrolytic activity. An exchange of H130 with tryptophan generated variants matching this residue as found in *Is*PETase and PHL4-6[4]. This mutation decreases both thermal stability (−4.4 °C) (Fig. 3d) and activity (15% of PET-hydrolytic activity after 8 h at 70 °C) (Fig. 3a) in comparison to PHL7 WT. We therefore assume that H130 is an adaptation allowing the performance of the enzyme at elevated temperatures. This is in line with the results of a comprehensive study on enhancing the thermostability of the mesophilic *Is*PETase[28]. By exchanging W159 in *Is*PETase (the structural equivalent of H130 in PHL7) to histidine, the melting temperature could indeed be increased by 8.5 °C. However, the PET-hydrolytic activity at 37 °C decreased as

well by this mutation. Therefore, we conclude that tryptophan in this position ensures activity at ambient temperatures (which is the case for *Is*PETase WT) while histidine is a prerequisite for activity at elevated temperatures (e.g., for PHL7, LCC and *Tf*Cut2).

Exchanging L210 to phenylalanine, which is a strictly conserved amino acid in this position in all type I polyester hydrolases, decreased the activity of PHL7 by about 50% (Fig. 3a, c). Type II polyester hydrolases active at ambient temperature such as PE-H (Y250) and *Is*PETase (S238) differ in this position as well. We conclude that both the size and the hydrophobicity of this residue influence the catalytic activity of the enzymes. Smaller amino acids (A, I, V, T, S) in position 210 generally increased the activity of PHL7 by 14% (L210S, 8 h) to 39% (L210T, 8 h), while the larger residue F decreased the activity by 32% (8 h) in comparison to the WT enzyme (Fig. 3a). On the other hand, the hydrophobicity of position 210 did not influence the activity of the enzyme significantly. We found that both hydrophobic (A, I, V) and hydrophilic (S, T) amino acid residues slightly increased the activity. These results are in accordance with a previous study on the polyester hydrolase *Tf*Cut2[49]. An exchange of the structurally equivalent F209 by small amino acids increased the PET hydrolysis rate twofold (F209S) to threefold (F209A). Bulkier hydrophobic amino acids such as leucine and tryptophan retained (F209L) or slightly increased (F209W) the activity of the enzyme. A similar behavior has been described for LCC, where an exchange of the corresponding residue F243 with I or W increased its PET-hydrolytic activity[5]. An exchange of the equivalent position in *Is*PETase by alanine (S238A) also supported this notion as it decreased BHET-hydrolytic activity by only 10%[22]. This mutation also increased the PET-hydrolytic activity and changed *Is*PETase from preferring *gauche* (WT) to *trans* PET conformations[48]. Amorphous PET predominantly (although not exclusively) contains *gauche* EG moieties, while the polymer chains in crystalline PET are in *trans* conformation[47,50]. On the other hand, modification of both adjacent residues in *Is*PETase subsite II (W159H/S238F) increased the activity slightly at 30 °C and also shifted the formation of hydrolysis products from TPA to MHET[24]. However, it is not possible to assign these improved enzymatic properties solely to either W159H or S238F since the authors did not present activity assays of the respective single mutants. It is therefore likely that W159H overcompensated negative effects of the bulky S238F exchange and that the PET-hydrolytic activity increased synergistically.

## Energetic contribution of L210 and its variants to substrate binding

We hypothesize that the stronger substrate binding of the F210 variant is responsible for a reduced catalytic activity by hindering the substrate or product release. We investigated the inhibition effect of MHET in the PHL7 WT and PHL7 L210F (Supplementary Fig. 6a,b). Despite the fact that MHET should solely bind to subsite I, we expected that variant L210F may be susceptible to inhibition by MHET. However, inverse Michaelis-Menten kinetics[51,52] with 20 mM MHET showed no clear difference in the reduction of the reaction rates of the WT and the variant L210F (Supplementary Fig. 6). When the PET surface becomes saturated with the enzyme, MHET may only have a negligible impact on the reaction rate. It has been previously shown that MHET as a product of PET degradation is subsequently hydrolyzed to TPA by PHL7.

To further investigate the role of position 210 we carried out ligand docking with the Rosetta docking protocol to assess the binding of 1,2-ethylene-mono-terephthalate-mono (2-hydroxyethyl terephthalate) (EMT) to PHL7 variants. The overall binding energy of the L210F mutant was significantly stronger compared to the WT enzyme (Fig. 4a). To validate our results, we also performed docking with AutoDockVina[53] which showed good agreement with the poses generated by Rosetta3 (Supplementary Fig. 8). To check ligand stability at the active site we performed molecular dynamics simulations with AMBER20[54]. The MD results suggested that the enzyme-ligand systems were stable as the

ligand remained in the active site during the simulation time in all variants except for replica 1 in the L210F variant (Supplementary Fig. 9 to 11). The overall binding energy of the substrate EMT was significantly stronger for the L210F mutant compared to the WT enzyme (Fig. 4a).

At the same time, the observed binding energy for selected variants was mirrored in their PET-hydrolytic activity (Figs. 3a, c and 4). We employed quantum chemical calculations based on the density functional theory (DFT) describing the electronic system by quantum mechanics (QM) to further investigate the interaction between the active site and EMT. The observed trends in the QM interaction energy ($E_{Int}$) were in line with the Rosetta predictions, since phenylalanine at position 210 resulted in an interaction energy 5.8 kcal mol$^{-1}$ stronger than in the WT (Fig. 4b). Conversely, threonine at that position resulted in a weakening of the binding energy by 3.9 kcal mol$^{-1}$ compared to the WT. The predicted per residue energy decomposition suggested that only phenylalanine at position 210 bound to the EMT ligand while the aliphatic side chains of L and T did not contribute (Fig. 4c). Mutagenesis of L210 did not significantly influence the energetic contribution of other residues to binding. Upon visual inspection we identified a wider distribution of binding poses for the second moiety of EMT binding to subsite II. Therefore, we calculated the conformational diversity of each phenylene ring structure (Fig. 4d) based on the previous observation that the phenylene units in PET are more mobile compared to the EG units[47]. We observed a wider distribution of RMSD values for TPA bound to subsite II in the three variants in comparison to TPA bound to subsite I. The QM calculations supported the insights gained from the molecular docking experiments since the binding energy of residue 210 decreased from threonine (−0.65 kcal/mol) to leucine (−3.15 kcal/mol) to phenylalanine (−5.48 kcal/mol). This trend matched the order of catalytic activity of PHL7 WT and L210 variants, indicating that a more favorable interaction energy at position 210 results in a decrease in the catalytic activity.

We therefore suggest that subsite I is the main binding contributor by interacting with one PET moiety to achieve a productive conformation at the catalytic serine. Subsite II could predominantly contribute to the initial substrate binding and guidance of the PET chain towards the active site (Supplementary Fig. 12). We speculate that subsite II may thus act as a guiding channel with rather loose interactions. Thus, binding of a single or two PET moieties may be sufficient to describe the mechanism. Subsite II may facilitate the enzyme to hold contact to its substrate and to transfer it towards subsite I. This model can sufficiently explain why MHET is formed as a main product, while release of TPA may predominantly occur as a subsequent reaction. Still, further in vitro and in silico experiments are required to demonstrate the proposed mechanism.

The metagenome-derived polyester hydrolase PHL7 demonstrates to be a useful tool for biological PET recycling. In this work we studied the binding of its product and generated variants with enhanced PET hydrolyzing activity and thermal stability. This study provides a structural basis for a subsequent systematic further enhancement of the catalytic activity of polyester hydrolases. We showed that PHL7 interacts with the TPA moiety of its substrate in a lock-and-key mechanism rather than an induced fit, similar to LCC and different from *Is*PETase. Mutating amino acids of the active site resulted in PHL7 variants with changed PET-hydrolytic activity and thermal stability, including double-gain mutants with higher activities and melting points compared to the WT (Supplementary Fig. 7). The single mutants L93F, Q95Y, L210T (or substitutions by other aliphatic amino acids) and D233K showed these improved properties. The presence of phenylalanine in position L210 decreased PET-hydrolytic activity by half due to a more favorable substrate binding energy contribution of this residue. Both a substantial thermal stability and PET-hydrolytic activity are prerequisites for the successful application of polyester hydrolases in biocatalytic PET degradation and recycling processes.

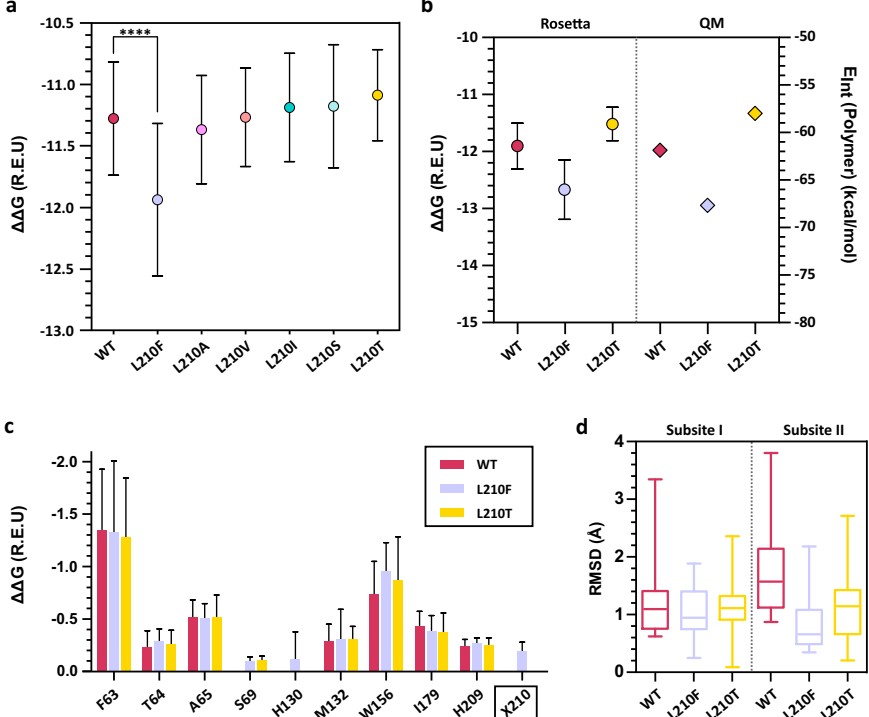

**Fig. 4 | Rosetta ligand docking and quantum chemical calculations based on density functional theory (DFT) of PHL7 and selected variants in position L210.** **a** Average Rosetta free binding energy ($\Delta\Delta G$) (mean ± SD) of the cluster that best represented the optimal binding poses of EMT ($n = 32$ independent calculations). The position of the terephthalic ring of EMT in the most populated cluster superimposes well with the terephthalic ring in the PHL7×TPA cocrystal structure. One-way Tuckey test was conducted to assess if the average $\Delta\Delta G$ of the most populated cluster was significant (****=$p < 0.0001$). Note that normal distribution was assumed as the number of data points did not allow for a normality test. **b** Comparison of Rosetta's average $\Delta\Delta G$ (left) (mean ± SD for $n = 10$ independent calculations) and of interaction energies $E_{Int}$ between EMT and the protein from DFT calculations (right, $n = 1$). Regarding the first, we report the mean ± SD interface_delta_X score of the 10 best scoring poses for the most populated cluster. **c** Per-residue binding energy (Rosetta $\Delta\Delta G$) for PHL7 WT and L210 mutants suggests that only Phe at position 210 but not Leu or Thr make a contribution to EMT binding (mean ± SD) for $n = 32$ independent experiments. **d** RMSD of the terephthalic rings of EMT binding to subsite I and II, respectively. For each system (WT, L210F and L210T), we selected the most populated cluster and calculated the RMSD of each terephthalic ring using the lowest energy pose as reference. Box-and-whisker plots with center line representing the median, boxes drawn from the lower to the upper quartile, and whiskers drawn between the lowest and highest RMSD values within 1.5 interquartile range of the lower and upper quartile ($n = 32$ independent calculations). REU Rosetta energy units.

## Methods

### Molecular cloning, protein expression and purification

PHL7 was expressed and purified by a three-step protocol as previously described[4]. In short, the enzyme was recombinantly expressed in *E. coli* BL21 (DE3) in 1 L LB medium, induction at OD 0.6 with 0.1 mM IPTG, followed by 16 h postinduction at 18 °C and 60 rpm. 50 mM Na-phosphate buffer, 200 mM NaCl, pH 7.4 was used as extraction, purification and storage buffer. Firstly, a Ni-NTA chromatography (5 mL HisTrap FF column (Cytiva, Marlborough, MA, USA), 20 mM and 40 mM imidazole washing steps and 250 mM imidazole elution step), followed by heat precipitation (55 °C for 15 min) and finally a size exclusion chromatography (Hiload 26/60 Superdex 200 pg column (Cytiva, Marlborough, MA, USA)) was applied to purify the protein variants to homogeneity. The proteins were stored at 4 °C. Protein concentration was determined by Bradford/BSA assay according to the suppliers manual (ROTI®Quant, Carl Roth GmbH+Co. KG (Karlsruhe, Germany)) and analyzed with a Synergy MX plate reader and Gen5 control software (BioTek Instruments, Inc., Winooski, VT, USA) at 450 and 590 nm.

### Site-directed mutagenesis

Mutations were introduced by site-directed mutagenesis. Primer, gene and vector sequences are included in the Source Data file. The PCR product was subsequently treated with DpnI according to the supplier's manual (NEB, Ipswich, MA, USA). *E. coli* XL10-Gold (Agilent

Technologies, Santa Clara, CA, USA) was transformed using 10 μL of the DpnI digested sample and plasmids were subsequently extracted following the standard protocol of the Monarch plasmid Miniprep kit (NEB, Ipswich, MA, USA). Mutations were confirmed by Sanger-sequencing (Microsynth Seqlab GmbH, Göttingen, Germany).

### Melting point analysis

Protein melting temperatures ($T_m$) were determined by nano differential scanning fluorimetry (nanoDSF) with a Prometheus NT.48 and control software (Nanotemper Technologies, Munich, Germany). Purified enzyme fractions with a concentration of 0.15 mg ml⁻¹ in sample buffer were analyzed. A heating ramp from 20 °C to 95 °C with a slope of 1 °C min⁻¹ was used. Mean values ± SD for $n = 3$ are shown.

### Gravimetric analysis of enzymatic PET hydrolysis

Amorphous PET (G-PET) film (Goodfellow GmbH, Bad Nauheim, Germany) coupons of $3 \times 0.5$ cm (about 45 mg) were used as substrate with an enzyme concentration of 0.55 mg$_{enzyme}$ g$_{PET}$⁻¹ in a total volume of 1.8 mL (not taking the PET film into account). The film weight loss was gravimetrically determined after 4 h, 8 h, and 16 h of reaction at 70 °C and 700 rpm in 1 M $K_2HPO_4$-NaOH, pH 8. The PET films were rinsed before and after the treatment with a sequence of water, 0.5% SDS, water and 70% EtOH. The reaction temperature was adjusted for enzyme variants with decreased thermal stability as depicted in the results section.

## Impedimetric determination of enzymatic PET degradation

The degradation of G-PET was monitored by the change in capacitive resistance of an approximately 225 µm thick PET film using impedance spectroscopy as described previously[55]. In short, a small piece of rectangular PET film (4.3 × 9.1 mm) was used as the thinnest material barrier in a divided cell setup (two reaction chambers) with a platinum electrode on each side. Each reaction chamber was filled with 1 M $K_2HPO_4$ (pH 7.8) and 500 nM enzyme (13.9 µg/mL), and heated up to 70 °C in a thermocycler, unless stated otherwise. For the inhibitor study, a fresh 40 mM stock solution of MHET was prepared in 2 M $K_2HPO_4$·NaOH (pH 8). Impedance was measured every minute using an ISX-3v2 high-precision impedance analyzer (Sciospec Scientific Instruments GmbH, Bennewitz, Germany) from 500 Hz to 1 MHz (41 frequency points) with a signal amplitude of 100 mV, controlled by self-developed measurement and automation software (IMA-Tadvanced). Impedance raw data were fitted with a MATLAB script using the simplified Randles equivalent circuit model to extract capacitance values. Change in capacitance was translated to a change in PET film thickness using a relative permittivity $\varepsilon_r$ of 3.3 and the starting thickness, which was measured mechanically (typically 225 µm). The enzymatic degradation rate was determined from the change in apparent layer thickness in the period between 5 h and 10 h by linear regression. Statistical significance was analyzed with one-way ANOVA and Tukey posthoc tests, considering *$P < 0.05$ as significant, **$P < 0.01$ as very significant, and ***$P < 0.001$ as extremely significant.

## Protein crystallization and structure determination of PHL7 S131A

The inactive PHL7 variant S131A was transferred into crystallization buffer (10 mM Tris, pH 7.4, 10 mM NaCl) and subjected to hanging drop crystallization. The crystallization of the wild type (WT) PHL7 has been described previously[4] and the same reservoir composition was utilized for cocrystallization of PHL7×TPA. Therefore, PHL7 S131A was concentrated to 8.7 mg/mL, and 1 µL of protein was mixed with 1 µL of reservoir solution. PHL7×TPA cocrystals were grown in a reservoir containing 100 mM Na₃-citrate pH 5.6, 20% (w/v) PEG 4000, 5% (v/v) 2-propanol, supplemented with 5% (v/v) TPA by adding a stock solution of 500 mM TPA in DMSO. PHL7×$Mg^{2+}$ crystals were grown in 50 mM Tris pH 8.5, 100 mM KCl, 30% (v/v) PEG 400 and 10 mM $MgCl_2$, supplemented with 5% (v/v) BHET by adding a stock solution of 500 mM BHET in DMSO. The drops were equilibrated against 500 µL of reservoir at 19 °C. Crystals grew after 1 to 5 days and were mounted and flash frozen in liquid nitrogen without further cryoprotection.

Diffraction experiments were conducted on a Synergy R diffractometer with a HyPix-6000HE detector (Rigaku, Tokyo, Japan). The datasets were indexed, integrated and reduced in the CrysAlis(Pro) software suite supplied by Rigaku[56]. The data were scaled and merged in AIMLESS[57] and the phase problem was solved by molecular replacement utilizing the apo structure of PHL7 (PDB ID: 7NEI). Refinement was conducted in PHENIX[58] and model building in COOT[59]. TPA restraints were calculated using the GRADE webserver by Global Phasing[60] and modified manually (see results section). Weak ice rings due to the lack of cryo protection impede the quality of the datasets. Molecular analyses and graphics were prepared using UCSF Chimera[61]. The statistics of data collection and refinement are summarized in Supplementary Table 2.

## Molecular docking and statistical analysis

To further explore the binding poses of PET into the active site of PHL7, the soluble substrate analog 1,2-ethylene-mono-terephthalate-mono (2-hydroxyethyl terephthalate) (EMT) was used for molecular docking with Rosetta3[23,62,63]. Molecular docking was also performed to further analyze the binding mode of the phenylene ring in the active site of the enzymes. The conformational diversity of the ligands was represented by creating 225 conformers for EMT using the confab package of Open Babel[64]. The conformers were selected based on an energy cutoff of 50 kcal mol⁻¹ and an RMSD cutoff of 1.3 Å for EMT. For the docking procedure, 40,000 enzyme-substrate and enzyme-product complex structures were generated with a custom Rosetta3 XML script enabling backbone and side-chain flexibility as previously described[23].

The top 50 docking poses for all enzyme-ligand pairs were subjected to pairwise ligand root mean square deviation (RMSD) calculations using DockRMSD to generate a distance matrix for unsupervised exploratory hierarchical clustering analysis. The ligand RMSD matrix was employed to build an Euclidean distance matrix using the dist() function built in R stats package and the Ward.D2 method was used to cluster the distance matrix[65]. We employed fviz_cluster function included in factoextra[66] and ggplot2[67] for visualization of the clustering results. The cluster that best represented the optimal binding poses of EMT into the active sites of PHL7 variants was determined by visually assessing that the phenylene ring of the molecules occupied the same site as the phenylene ring of the TPA molecule present in the cocrystal structure PHL7xTPA. The complexes from these selected clusters were subjected to decomposition of the per-residue energetic contributions to ligand binding using Rosetta3 ddG mover[68].

Previous solid state NMR studies on PET films[47] reported that the phenylene ring engaged a higher flexibility in comparison to the ethylene glycol. Based on this evidence we aimed to quantify the conformational diversity for each subsite by calculating ring-RMSD.

One-way ANOVA with post-hoc Tukey test was performed using Graphpad Prism to assess statistical significance of each group with respect to the WT.

## Quantum mechanical calculations of selected L210 variants

All quantum mechanical calculations were realized with the Gaussian 16 software suite[69]. We started from the best docking models for WT, L210T and L210F. To model the active site, we included residues 63 to 65 and residues 130 to 132 with corresponding parts of the protein backbone and selected sidechains of the following residues: S69, L93, Q95, W156, I179, H209, L/T/F210. In case of the former, we cut through the protein backbone between the carbon-carbon bonds of the preceding glycines and of the last-mentioned residue. When only including the sidechains, we cut through the carbon-carbon bonds between $C_\alpha$ and $C_\beta$. In both cases, hydrogen atoms were added to saturate the truncated bonds. We only optimized the position of all hydrogen atoms employing the B3LYP functional[70–72] together with Grimme dispersion correction with Becke-Johnson damping (GD3BJ)[73,74] and def2-SVP basis set[75]. Subsequently, we performed single point calculations to evaluate the reported interaction energies. For this, we determined the total energy of the whole active site as well as of residue 210 (X210) and the active site without that residue. The interaction energy for the sidechain of residue 210 was then obtained as:

$$E_{Int}(X210) = E(ActiveSite) - \big[E(X210) + E(ActiveSiteWithoutX210)\big] \tag{1}$$

An analogous approach was employed to evaluate the interaction energies of the polymer with the active site.

## Reporting summary

Further information on research design is available in the Nature Portfolio Reporting Summary linked to this article.

## Data availability

The data generated in this study are available within the article and the supplementary information. Crystallographic structures have been deposited at the Protein Data Bank (https://www.rcsb.org/) under accession numbers 8BRA and 8BRB. Further data from our Rosetta simulations and DFT calculations are available upon request. Source data are provided with this paper.

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

## Acknowledgements

We gratefully acknowledge funding from the European Union's Horizon 2020 research and innovation program under grant agreement no. 887913 (ENZYCLE). PF acknowledges the Deutsche Bundesstiftung Umwelt (DBU, Grant no. 20018/565) for financial support. RF and DK received funding by the Federal Ministry for Economic Affairs and Energy based on a resolution of the German Bundestag (BMWi, STARK program Grant 46SKD023X). The DFT calculations were performed on resources provided by the Leipzig University Computing Center. We thank the MX Laboratory at the Helmholtz-Zentrum Berlin (BESSY II) and the EMBL beamlines of the DESY synchrotron in Hamburg for synchrotron beamtime. Funded by the Open Access Publishing Fund of Leipzig University supported by the German Research Foundation within the program Open Access Publication Funding.

## Author contributions

W.Z. and J.M. conceived the general project idea. C.S., K.R., and N.S. designed the experiments. K.R. and N.S. performed the crystallographic experiments. P.B.S., Z.Z., E.F., Y.L., and P.F. performed the mutagenesis, enzyme production and characterization experiments. R.F. and D.K. performed the impedance-spectroscopic analysis. F.E., G.K., and P.B.S. performed docking experiments. C.W. performed the DFT calculations. K.R. wrote the manuscript with support from C.S., W.Z., and N.S. C.S. supervised the project. All authors provided critical feedback and contributed to the writing of the manuscript.

## Funding

## Competing interests

The authors declare no competing interests.
