## [Peer Review File · Nature Communications]

REVIEWER COMMENTS

Reviewer #1 (Remarks to the Author):

The manuscript has no page number, which gives trouble to designate the exact positions in writing comments. I have tentatively given page numbers starting from the title page. The paper elucidates the distinct differences between thermophilic PET hydrolases and mesophilic PET hydrolases, based on the structural analyses of both groups. Even if the whole structures of two groups are homologous to each other, the minute structural differences cause the fatal differences in their activities and thermostabilities, as described in this paper, which endorses the conclusion that the thermophilic PET hydrolase group is better than the mesophilic PET hydrolase group and PHL7 mutants are the best among the thermophilic PET hydrolases so far reported.

It is no doubt that the Results and Discussion and deduced Conclusions are reasonable in general. The paper surely promotes the research on practical PET hydrolases for bio-recycling of PET. However, unfortunately, there are some problems to be improved/revised in the paper, as described below.

1. In this paper, the principal difference between the two groups is elucidated by the wider open in IsPETase due to S214 (wobbling of W185) (section 1 in Results and Discussion). On the other hand, da Costa et al (Proteins 2021:1-13) suggested that the β 1- β 2 loop (causing more flexibility in IsPETase) is to be mutated for improved thermostability. At least their paper is to be cited and discussed.

2. Page 8, the second section in Results and Discussion: Mg or Ca ions are more efficient for activity and thermostability than Na ions. However, in the previous paper (ref 4), it was described that the binding site for Na (E148, D233, and F230) appears well suited to coordinate Ca ions. As the Ca-binding site is shown in this paper, the previous expectation was betrayed. What do you think about two meta-binding sites? Does the coexistence of Na and Mg enhance activity/thermostability? Mutation at D233 to K233 increased the activity slightly. For what reason? In addition, D246 corresponds to the amino acid (E) related to Ca-binding in TfCut2 and Cut190, but D247 and E13 (Is this correct? Not E11 from Fig. S1 in ref. 4?) are quite different from the amino acids related to Ca-binding in TfCut2 and Cut190 (their crystallization results have been solved). However, the sequences are homologous to each other. What causes these differences? A disulfide bond was introduced in LCC ICCG mutant to counteract the effect of Ca ion at the corresponding position to the Ca-binding site in TfCut2 and Cut190. As the impact of metal ions is essential for PET hydrolases, more discussion is required in detail.

From the context of this section, the section title is inappropriate and therefore is expected to be revised such as Metal-binding sites for Na and Mg/Ca ions.

3. Page 9, the sentences starting from Chen et al.---- (Fig.3b): This section is not related to the section title (Functional analysis of subsite I residue -----). Move to an appropriate section.

4. Page 10, section 4: Fig. 1b in the third line must be Fig. 1c. Regarding the effect of L93F and Q95Y, why not describe the results of LCC by Tournier et al.? In addition, what was the last sentence (The two residues H185 and F189---) based on? Cite the corresponding result or references. I wonder if this section could be combined with section 3, as section 4 is related to subsite 1 (section 3).

5. Page 11, section 5, the first paragraph: Kawabata et al. (J. Biosci. Bioeng. 124: 28-35, 2016) published the 3D docking structure of Cut190 with model compounds of PET and

suggested that 3-5 monomer units such as TET(ET) fill the active site and no more units are accommodated. As Cut190 shares the protein structure homologous to other thermophilic PET hydrolases, this might be useful for discussion here.

6. Page 12: Was EMT defined in the above text? If not, please define it here.

7. Page 13, line 2: How are Fig. 2a and 2c related to this sentence? Probably you must mean Fig. 3.

8. Supplementary information: References are not cited in the Tables and Figures. It is better to cite references exactly at the corresponding positions.

Reviewer #2 (Remarks to the Author):

This manuscript describes a comprehensive analysis and comparison of three PET hydrolases, namely PHL7 previously described by the authors, the mesophilic *IsPETase* and the thermophilic enzyme LCC. The crystal structure of PHL7 was solved as a complex with the hydrolysis product terephthalic acid (TPA) and substrate binding sites were identified. Furthermore, differences between these PETases were examined with regard to thermostability and activity.

General remarks:

(1) This is a carefully conducted study which presents interesting results.

(2) For the readers, it is often difficult to follow the rationale behind making and analyzing the described mutations. The authors present an overview of the mutations in table S1. Nevertheless, it would be helpful to have them schematically presented in a figure showing, e.g. the amino acid sequences and 3D structures and indicating which residues belong to which enzyme and affect which property.

(3) Figure S6 summarizes the most important results. It should be presented in the main manuscript.

Specific comments:

Lines 28-29 and line 338: This statement needs to be validated. The authors did not compare the activities and stabilities of the mentioned PETases under the same experimental conditions.

Table 1: The numbers in column: ref (reaction temperature) should be explained.

Fig. 3: In panel (a), the temperature should be mentioned. In panel (b), the time should be mentioned. Why are three variants shown in the left panel, and five variants in the right panel? In panel (d), variant S131A is shown which is missing in panel (a).

Line 267: The role of aa L210 in PHL7 and its homologues is known and well documented in the literature. It would therefore be highly interesting to construct and comparatively analyze a complete site saturation library at this position.

Line 291: In this paragraph, it should be discussed how the authors think that the PET dimer is bound.

Line 301: Why are just these variants analyzed? The abbreviation EMT should be explained

at the beginning of the manuscript, not at the end (line 417).

Fig. 4: This figure is mislabeled as Fig. 1. In panel (c): why is X210 printed in red as the WT?

Reviewer #3 (Remarks to the Author):

In this study, Richter, P. et al. have employed the molecular docking procedure using the Rosetta3 package to explore binding poses of PET into the active site of PHL7. The binding energies of the top poses provided by the docking procedure were compared with the energies computed using single point DFT calculations. Overall, calculations are performed competently and results are described clearly. The authors should consider the following points and revise the manuscript.

(1) The authors have placed a lot of confidence on the structures of the enzyme-substrate and enzyme-product complexes provided by Rosetta3. It is well known that docking protocols are not very reliable and quite often inclusion of dynamics completely alters the structures provided by them. The authors should run molecular dynamics (MD) simulations at least on some of the most promising poses for both WT- and the mutant forms of the enzyme to test their stability. That would make their results more credible and further strengthen the manuscript.

(2) They should also test the accuracy of the poses provided by Rosetta3 by performing docking with at least one more software.

We thank the reviewers for their very valuable and helpful comments. We think that the manuscript has been significantly improved with the help of their excellent suggestions. We have submitted a revised version (page numbers and lines refer to this revised version). Further we have added a track-changes docx document showing all changes that have been made during the revision process.

To incorporate the reviewers' suggestions, the revised manuscript now contains supplementing datasets, which are presented in a number of additional tables and figures.

Additional Tables and Figures:

Supplementary Table 3. Rank and score of Autodock Vina predictions and RMSD to the Rosetta predicted structure for each variant.

Supplementary Table 4. Score of Autodock Vina for the nine docking structures predicted for each variant. The structure that most closely resembles the Rosetta prediction is marked in yellow.

Supplementary Figure 1: Overview of the sites mutated in this study and origin of the chosen replacement residues.

(original Supplementary Figure 1 was moved and is now Supplementary Figure 5)

Supplementary Figure 8: Results of Autodock Vina control docking calculations.

Supplementary Figure 9: Overview of the largest molecule clusters of the 50 best energy docking models of EMT for each L210X variant tested.

Supplementary Figure 10. Docking poses of EMT in PHL7 overlaid with the crystallized TPA molecule in PHL7.

Supplementary Figure 11. MD simulation results for PHL7-EMT docking model.

Supplementary Figure 12. Predicted binding mode of PHL7 with a 2-HE(MHET)₄ generated using the DiffDock deep learning model.

Reviewer 1:

Request 1/1:

“In this paper, the principal difference between the two groups is elucidated by the wider open in IsPETase due to S214 (wobbling of W185) (section 1 in Results and Discussion). On the other hand, da Costa et al (Proteins 2021:1-13) suggested that the β 1- β 2 loop (causing more flexibility in IsPETase) is to be mutated for improved thermostability. At least their paper is to be cited and discussed.”

Response 1/1:

We thank the reviewer for the valuable comment. We included the following information to the discussion (p. 7, line 138): “Besides the conformational change of loop β 7- α 5 upon substrate binding in IsPETase, MD-simulations from da Costa et al. further identified loop β 1- β 2 to be very flexible.”

Request 1/2:

*“Page 8, the second section in Results and Discussion: Mg or Ca ions are more efficient for activity and thermostability than Na ions. However, in the previous paper (ref 4), it was described that the binding site for Na (E148, D233, and F230) appears well suited to coordinate Ca ions. As the Ca-binding site is shown in this paper, the previous expectation was betrayed. **What do you think about two metal-binding sites? Does the coexistence of Na and Mg enhance activity/thermostability? Mutation at D233 to K233 increased the activity slightly. For what reason?** In addition, D246 corresponds to the amino acid (E) related to Ca-binding in TfCut2 and Cut190, but D247 and E13 (Is this correct? Not E11 from Fig. S1 in ref. 4?) are quite different from the amino acids related to Ca-binding in TfCut2 and Cut190 (their crystallization results have been solved). However, the sequences are homologous to each other. **What causes these differences?** A disulfide bond was introduced in LCC ICCG mutant to counteract the effect of Ca ion at the corresponding position to the Ca-binding site in TfCut2 and Cut190. As the impact of metal ions is essential for PET hydrolases, more discussion is required in detail.”*

General response 1/2:

Based on the reviewer's comments, we have substantially revised this section in the manuscript text, now entitled "Metal binding sites". It is clearer now that two metal binding sites have been identified on the surface of PHL7, which are attractive targets for further protein stabilization engineering. The binding sites have been interpreted in the crystal structure refinement process as Na⁺ (previous study) and Mg²⁺ (this paper), based on coordination distances and the high concentrations of these ions in the crystallization buffers. We agree with the reviewer, that it would be of interest to study possible additive effects of the binding of different metal ions. However, we think that this requires a systematic study, including also an investigation of the general effects of increasing salt concentrations on protein stability as well as the design of systematic mutagenesis experiments. The additivity of activation by Ca²⁺ and Na⁺ has not been tested by us, but we found that Na⁺ increases PHL7 stability even in the presence of high phosphate concentration, which is a strong stabilizing anion according to the Hofmeister series. Based on the high surface and solvent exposure and the spatial distance of 26 Å, we assume that the two binding sites may result in independent and thus additive effects on protein stability. These points have now been addressed in the manuscript text.

1. ***“Does the coexistence of Na and Mg enhance activity/thermostability?”***

Response 1/2: We have observed stabilizing effects, but no increase in activity. Ca²⁺ showed the strongest stabilization effect, but the coexistence of Na and Mg has not been tested.

2. ***“Mutation at D233 to K233 increased the activity slightly. For what reason?”***

Response 1/2: Concerning the effects of the D233K mutation, we propose the formation of an electrostatic interaction between the positively charged D233K and the negatively charged E148. This direct interaction between two oppositely charged residues would make the presence of a

mediating divalent cation obsolete in order to stabilize the protein structure. The slightly increased thermal stability could in turn increase the PET-hydrolytic activity at 70 °C due to an increased thermal half-life, although this hypothesis was not proven by us. In our manuscript, we only report on the observed effect of this mutation.

- 3. *“In addition, D246 corresponds to the amino acid (E) related to Ca-binding in TfCut2 and Cut190, but D247 and E13 (Is this correct? Not E11 from Fig. S1 in ref. 4?) are quite different from the amino acids related to Ca-binding in TfCut2 and Cut190 (their crystallization results have been solved). However, the sequences are homologous to each other. What causes these differences?”***

Response 1/2: None of the Ca²⁺- or Mg²⁺-binding sites of TfCut2 or Thc_Cut2 (Ribitsch et al., 2017) resemble the novel Mg²⁺-binding site of PHL7 identified in this study. However, multiple metal-binding sites were identified in Cut190, of which sites 1-3 influence both the thermal stability and the activity of the enzyme and are usually occupied by Ca²⁺ (Numoto et al., 2018). The additionally identified binding sites 4 and 5 are occupied by Zn²⁺ in two cocrystal structures (PDB IDs: 5ZRQ, 5ZRR), of which Zn²⁺ bound to site 4 interacts with E57 and D292. These residues correspond to residues E13 and D247 in PHL7, which are part of the Mg²⁺-binding site comprised of E13, D246 and D247. However, D246 of PHL7 is replaced by T291 in Cut190. No functional relevance was reported for either of the Zn²⁺-binding sites 4 and 5 of Cut190. We added the following information to the section "Metal binding sites": Page 8, line 177: "This Mg²⁺-binding site of PHL7 resembles a Zn²⁺-binding site identified in the thermophilic homolog Cut190 (binding site 4), which was described as a site unrelated to PET-hydrolytic activity."

(The sequence alignment of Figure S1 of reference 4 starts at a conserved region of the aligned proteins, therefore the residue appearing at position E11 is in fact E13.)

- 4. *“A disulfide bond was introduced in LCC ICCG mutant to counteract the effect of Ca ion at the corresponding position to the Ca-binding site in TfCut2 and Cut190.”***

Response 1/2: We added the following sentence in the main text together with corresponding references to inform about successful experiments to increase thermostability by replacing metal binding sites with disulfide bridges in polyester hydrolases: Page 8, line 171: "Further reports also describe increased thermostability by introducing a disulfide bridge in TfCut2, Cut190 or LCC at positions of metal binding sites". These sites do not correspond to the two observed metal binding sites of PHL7, however.

- 5. *“As the impact of metal ions is essential for PET hydrolases, more discussion is required in detail.”***

Response 1/2: We agree with the reviewer about the importance and impact of metal binding sites for PET hydrolase stability and activity. In addition to the other revisions described here, we added information and citations for studies demonstrating metal activation of binding sites (p. 8, line 177): "This Mg²⁺-binding site of PHL7 resembles a Zn²⁺-binding site identified in the thermophilic homolog Cut190 (binding site 4), which was described as a site unrelated to PET-

hydrolytic activity¹². In summary, two metal binding sites have been identified at the surface of PHL7. Similar to previous studies on other polyester hydrolases, these may as well serve as mutagenesis targets in upcoming experiments in order to increase the thermal stability of PHL7.”. For Cut190 the influence of Ca²⁺ on the catalytic mechanism for PET hydrolysis was demonstrated in multiple studies (Kawai et al., 2014; Numoto et al., 2018; Oda et al., 2018; Emori et al., 2021 and others). Other polyester hydrolases such as IsPETase have not yet been characterized as metal ion-dependent yet. For most other polyester hydrolases (LCC, TfCut2 etc.), the presence of divalent cations such as Ca²⁺ or Mg²⁺ increases their thermal stability, but does not increase their PET-hydrolytic activity. Besides Cut190, no polyester hydrolase is known to depend on the presence of Ca²⁺ in order to hydrolyze its substrate. A more detailed discussion should take place in the context of further studies addressing the thermostability of these enzymes.

6. ***“From the context of this section, the section title is inappropriate and therefore is expected to be revised such as Metal-binding sites for Na and Mg/Ca ions.”***

Response 1/2: The title of the chapter was changed to “Metal-binding sites”(p. 8,line 152).

We would like to emphasize a point regarding the metal-binding sites of PHL7.

PHL7 is most active in a 1 M phosphate buffer. Even the addition of small amounts of divalent metal ions such as Mg²⁺ or Ca²⁺ would lead to precipitation of their respective phosphate salts. We therefore could not investigate the influence of Mg²⁺ or Ca²⁺ on the PET-hydrolytic activity of PHL7 without changing the optimized buffer system.

Request 1/3:

"Page 9, the sentences starting from Chen et al.---- (Fig.3b): This section is not related to the section title (Functional analysis of subsite I residue -----). Move to an appropriate section."

Response 1/3:

The mentioned paragraph reports on amino acid modifications in the homologous polyester hydrolases IsPETase, TfCut2 and LCC corresponding to positions H185 and F189 in PHL7. Although these two residues are not shown in Figure 1a as the surface-forming residues in direct contact with the substrate, we classify both amino acids as subsite I residues since they directly interact with the residue corresponding to W156 in PHL7 (see also Fig. 1c which presents an overview of subsite I and II residues). The W156 residue is part of the π -stacking clamp of subsite I and therefore modifications of amino acids that are likely to directly influence the conformation of W156 will disrupt the structural integrity of subsite I. Therefore, we prefer to discuss these two residues in this section.

Request 1/4:

"Page 10, section 4: Fig. 1b in the third line must be Fig. 1c. Regarding the effect of L93F and Q95Y, why not describe the results of LCC by Tournier et al.? In addition, what was the last sentence (The two residues H185 and F189---) based on? Cite the corresponding result or references. I wonder if this section could be combined with section 3, as section 4 is related to subsite 1 (section 3)."

Response 1/4:

The two mentioned chapters 3 and 4 were combined into one. The title was changed to "Functional analysis and systematic mutagenesis of subsite I residues" (p.8, line 184). This chapter now covers a review of subsite I residues and systematic exchanges in homologous polyester hydrolases, and the results of our own mutagenesis study on PHL7. An introductory paragraph was added: *Page 8, line 185 to 190) "Subsite I of PHL7 is composed of five amino acids that directly interact with the substrate (F63, M132, W156, I179 and H209, see Fig. 1a and 1c), and four residues that can be classified as the second interaction sphere, meaning that they influence the conformation and flexibility of the aforementioned binding residues (L93, Q95, H185, F189). We compared the amino acid composition of subsite I of PHL7 with the homologous polyester hydrolases IsPETase, LCC and TfCut2 (both WT and engineered variants) and mutated selected residues to match them (Supplementary Fig. 1)."*

The chapter regarding the PHL7 mutations L93F and Q95Y was complemented with information on the activity of the respective LCC variants. Unfortunately, the original source does not specify exact activity values (only for Y127G (Q95G in PHL7)). We added the following sentence to the discussion. Page 11, line 248: "LCC variant Y127G showed an increased thermostability by 2.3°C, while the residual activity dropped down to 67%⁶."

The last sentence of this chapter, on the role of residues H185 and F189, was modified so that it compares our own thermal stability measurements with data from Chen et al. (Nature Catalysis, 4, 425–430 (2021)) who introduced the respective double mutation into the thermophilic polyester hydrolases TfCut2 and LCC. Page 11, line 260: "The two residues H185 and F189, adjacent to W156, are a prerequisite for thermal stability of PHL7 above 60 °C (Fig. 3d) , which is also reflected by data obtained on the respective double mutants for the thermophilic polyester hydrolases TfCut2 and LCC⁴²"

Request 1/5:

"Page 11, section 5, the first paragraph: Kawabata et al. (J. Biosci. Bioeng. 124: 28-35, 2016) published the 3D docking structure of Cut190 with model compounds of PET and suggested that 3-5 monomer units such as TET(ET) fill the active site and no more units are accommodated. As Cut190 shares the protein structure homologous to other thermophilic PET hydrolases, this might be useful for discussion here."

Response 1/5:

We thank the reviewer for this valuable suggestion. We have added the mentioned reference to our manuscript at p.11, line 268: "Potential binding of PET compounds with three to five monomer units has been described for Cut190, while mutational studies along the binding cleft showed most drastic effects for subsite I residues."

Request 1/6: "Page 12: Was EMT defined in the above text? If not, please define it here."

Response 1/6:

The abbreviation EMT was added upon its first reference in the text (p. 13 , line 325).

Request 1/7:

"Page 13, line 2: How are Fig. 2a and 2c related to this sentence? Probably you must mean Fig. 3."

Response 1/7:

We have corrected the reference to figure 3 instead of 2.

Request 1/8:

"Supplementary information: References are not cited in the Tables and Figures. It is better to cite references exactly at the corresponding positions."

Response 1/8:

We have removed the column "references" in Table S1 and added the references directly behind the referenced proteins. Thereby we changed the footnotes of Table S1 from numbers to letters. We added reference 2 into the caption of Figure S6.

Reviewer 2:

Request 2/1:

"For the readers, it is often difficult to follow the rationale behind making and analyzing the described mutations. The authors present an overview of the mutations in table S1. Nevertheless, it would be helpful to have them schematically presented in a figure showing, e.g. the amino acid sequences and 3D structures and indicating which residues belong to which enzyme and affect which property."

Response 2/1:

We have added a new figure (Supplementary Figure 1) that contains a multiple protein sequence alignment of the enzyme sequences from Supplementary Table 1 together with a model of the PHL7 substrate binding site. We have highlighted the binding site residues for subsite I (blue) and II (green), the catalytic triad (red) and marked the mutation sites (numbers in alignment, underlined in the model). Beyond, the model shows a proposed binding mode for EMT from docking experiments (dark grey) and the bound TPA (yellow) from the co-crystal structure.

Request 2/2:

"Figure S6 summarizes the most important results. It should be presented in the main manuscript."

Response 2/2

Figure S6 has not been placed in the main manuscript since the data is redundant with those shown in Figure 3a and d.

Request 2/3:

"Lines 28-29 and line 338: This statement needs to be validated. The authors did not compare the activities and stabilities of the mentioned PETases under the same experimental conditions."

Response 2/3:

We exchanged the last sentence in the abstract (p.2, lines 28-29) to the following sentence: "Variant L210T showed significantly higher activity, achieving a degradation rate of 20 $\mu\text{m h}^{-1}$ with amorphous PET films"

We exchanged the first sentence of the Conclusions (p.16, line 376) with the following sentence: "The metagenome-derived polyester hydrolase PHL7 demonstrates to be a useful tool for biological PET recycling."

Request 2/4:

"Table 1: The numbers in column: ref (reaction temperature) should be explained."

Response 2/4:

This column contains the references from which the optimal reaction temperature was extracted. The column "ref." (crystal structure) contains the references where the crystal structure of the respective protein was published. We have removed the "ref" columns and added the corresponding references directly behind the temperature values and PDB IDs.

Request 2/5:

"Fig. 3: In panel (a), the temperature should be mentioned. In panel (b), the time should be mentioned. Why are three variants shown in the left panel, and five variants in the right panel? In panel (d), variant S131A is shown which is missing in panel (a)."

Response 2/5:

We have added the information about the experiment temperature to the caption of Fig. 3a. The experiment time was added to the caption of Fig. 3b as well.

Panel (b) shows PET-hydrolytic activities of variants with strongly decreased thermal stabilities (see panel (d) of the same figure) at decreased temperatures in order to verify that the decreased activity of the variants is not a result of protein instability at 70 °C but due to a decreased PET-hydrolytic activity of the variant. In panel (d) we focus on the five most relevant variants and verify significant differences in their activities by impedance spectroscopy.

The inactive PHL7 variant S131A did not show any PET weight loss activity after 24 h which is why we omitted it from panel (a) but included it in the thermostability assay (panel d). We added this information in the figure caption.

Request 2/6:

"Line 267: The role of aa L210 in PHL7 and its homologues is known and well documented in the literature. It would therefore be highly interesting to construct and comparatively analyze a complete site saturation library at this position."

Response 2/6:

We generally agree with the reviewer's comment. We note that we were interested in finding the most active variant for position 210. By a rational design approach, we excluded several residues that are likely disturbing the catalytic activity or stability of the enzyme. Charged residues (D,E,K,R,H) likely disturb substrate interaction, the aromatic residue Y likely has a similar effect as F. We use A instead of G to test the effect of a small side chain (Ala-scan). P was excluded as it is likely distorting the conformation of the active site. We thereby focused on residues that are less likely to have a disturbing effect (thus we only allowed codons with T or C at the second position to cover all L210X mutations in one primer).

Request 2/7:

"Line 291: In this paragraph, it should be discussed how the authors think that the PET dimer is bound."

Response 2/7:

We have added additional figures (Supplementary Figure 1 and 12) where we show the proposed binding of EMT and an 2-HE(MHET)₄ (ETETETETE) molecule, respectively. We further added a paragraph at the end of the discussion that proposes a mechanism how PHL7 could process a PET chain. Page 15, Line 366 to 373: *"We therefore suggest that subsite I is the main binding contributor by interacting with one PET moiety to achieve a productive conformation at the catalytic serine. Subsite II could predominantly contribute to the initial binding of the PET chain and guidance of the PET chain towards the active site (Supplementary Figure 12). We speculate that subsite II may thus act as a guiding channel with rather loose interactions. Thus, binding of a single or two PET moieties may be sufficient to describe the mechanism. Subsite II may facilitate the enzyme to hold contact to its substrate and to transfer it towards subsite I. This model can sufficiently explain why MHET is formed as a main product, while release of TPA may predominantly occur as a subsequent reaction. Still, further wet lab and in silico experiments are required to demonstrate the proposed mechanism."*

Request 2/8:

"Line 301: Why are just these variants analyzed? The abbreviation EMT should be explained at the beginning of the manuscript, not at the end (line 417)."

Response 2/8:

We conducted Rosetta binding free energy calculations of EMT binding to PHL7 for all L210 variants which we also expressed and used for weight loss experiments. We further selected a set of variants for the QM calculations to support our hypothesis of the role of residue 210 for ligand binding. We could show that PHL7 with the aromatic phenylalanine in position 210 has a decreased PET-hydrolytic activity (Figure 3) while at the same time showing stronger binding to the substrate (Figure 4). This gives us a clear support for our hypothesis: the stronger the amino acid in position 210 interacts with the substrate, the lower is the PET-hydrolytic activity. The abbreviation EMT was added upon its first reference in the text (p. 13, line 325).

Request 2/9:

"Fig. 4: This figure is mislabeled as Fig. 1. In panel (c): why is X210 printed in red as the WT?"

Response 2/9:

The wrong label was corrected. The color coding of WT, L210F and L210T refers only to the color of the bars, not the color of the label. Only position L210 was exchanged in order to create figure 4c, but the Rosetta binding energy was calculated per residue. This is why the amino acid in position 210 is labeled as X210. Red bars refer to per-residue binding energies of the WT (L210), blue bars refer to per-residue binding energies of the L210F mutant, and yellow bars refer to per-residue binding energies of the L210T mutant. To clarify, we replaced the colored X210 label by a framed black label

Reviewer 3:

Request 3/1:

"The authors have placed a lot of confidence on the structures of the enzyme-substrate and enzyme-product complexes provided by Rosetta3. It is well known that docking protocols are not very reliable and quite often inclusion of dynamics completely alters the structures provided by them. The authors should run molecular dynamics (MD) simulations at least on some of the most promising poses for both WT- and the mutant forms of the enzyme to test their stability. That would make their results more credible and further strengthen the manuscript."

Response 3/1:

We agree with the reviewer's comment and, in order to validate the structural stability of the PHL7-EMT and PHL7-TPA complex models obtained from docking, we have performed additional molecular dynamics simulations.

We would like to remark that the RosettaLigand docking protocol we used differs from other deterministic docking protocols in at least two critical points: (1) the flexibility of the protein and ligand are defined at multiple stages of the protocol through sidechain rotamer trials and ligand conformer replacements, (2) with sufficient sampling, multiple conformations of a free energy basin can be obtained by Monte Carlo modeling, allowing to assess, at least in part, the ligand dynamics in the active site.

As explained in the manuscript, we used extensive sampling and we filtered and clustered the docked poses by their energy to select the most frequently occurring conformations close to the binding energy minima. We have included the detailed results of our docking and clustering protocol as a new supporting figure (Supplementary Figure 9). As can be seen, for each variant, the most populated cluster (cluster 1, 60-76% frequency) had the terephthalic ring in the same position and orientation as that observed in the crystal structure of PHL7-TPA (Supplementary Figure 9 and 10). Nevertheless, according to the calculated Rosetta score, other docking poses from less populated clusters exhibited similar or lower binding energies than the poses from the largest cluster 1. This suggests that in experimental conditions, additional EMT binding modes could occur. To further explore this observation, we performed molecular dynamics simulations using the lowest energy docking model of the largest cluster as starting structure for the WT and each L210X variant. We simulated each enzyme-ligand complex in four replicas of 100 ns at 298K (see Methods). Overall, the MD results suggested that the PHL7-EMT complex model was stable since the ligand remained in the active site in all variants for all replicas except for one replica (replica 1) in the L210F variant. In addition, we measured the distance between the γ -oxygen (OG) of the catalytic serine (S131) and each of the EMT oxygen atoms during the simulation. We consider a distance of 2.5-5 Å between the γ -oxygen of the serine and the EMT oxygen atoms as necessary for catalysis to happen. The results showed that this 2.5-5 Å distance range is maintained for almost the entire simulation time in all simulations with the exception of the L210F variant replicas 1 and 2 (Supplementary Figure 11).

Request 3/2:

"They should also test the accuracy of the poses provided by Rosetta3 by performing docking with at least one more software."

Response 3/2:

We have considered the reviewer's suggestion for validating our EMT docking results by using another docking method. We performed docking experiments using AutoDock Vina and found a good structural agreement with the Rosetta docking results for all variants in at least one out of nine AutoDock Vina predictions (Supplementary Figure 8). As highlighted in Supplementary Tables 3 and 4, for all L210X variants there was at least one model amongst the 9 top-scoring AutoDock predictions that closely resembled (RMSD < 3.0Å) the docking model obtained with RosettaLigand.

We have added the following sentences that refer to response 3/1 and 3/2 (p. 13, line 326 to 331): *"To validate our results, we also performed docking with AutoDock Vina which showed good agreement with the poses generated by Rosetta (Supplementary Fig. 8). To check ligand stability at the active site we also performed molecular dynamics simulations with AMBER20. The MD results suggested that the enzyme-ligand systems were stable as the ligand remained in the active site during the simulation time in all variants except for one replica (replica 1) in the L210F variant (Supplementary Fig. 9 to 11)."*

REVIEWERS' COMMENTS

Reviewer #1 (Remarks to the Author):

The revision is easier to follow the flow of contexts than the first version.

Still there are minor points to be modified, as shown below.

1. Line 193: The same effect---- is inappropriate, as IsPETase and LCC have Y at this position. Probably you would like to mean that mutation of Y to F and A in both enzymes have the same effect of F to Y mutation of PHL7.

2. Numbers of supplementary figures should be revised: Two figure-1s exist and the 2nd Figure 1 to Figure 3 should be Figures 2 to 4.

3. Line 234: Should be "Exchange of M161 or I208 in IsPETase--.

4. Lines 297-: In references 47 and Kawai et al. (AMB Express, 12: 134 (2022))), it is described that replacement of F with A and I remarkably decreased the expression of mutant enzymes, indicating that F is crucial to Cut190. This might be due to the different amino acids interacting with F in Cut190, thereby leading to different binding energy of F in PHL7 and Cut190.

5. Conclusions: Supplementary Fig. 7 does not include double-gain mutants. Why not them?

Reviewer #2 (Remarks to the Author):

The points raised previously were answered satisfactorily.

Minor points remaining are:

(1) Legend to Fig. S1: PDB no. for PHL7-TPA crystal structure is missing; probably not assigned yet?

(2) I find former Fig. S6 much easier to understand than Figs. 3a and d. If these figures are redundant, why are they presented in the main text and, additionally, as supplementary figure?

(3) It was a bit irritating that the authors mentioned in their rebuttal letter page and line numbers of the original manuscript rather than of the revised version which contains the additions/changes that they referred to.

Reviewer #3 (Remarks to the Author):

The authors have addressed all previously raised issues. There are no further concerns.

We thank the reviewers for their very valuable and helpful comments. We think that the manuscript has been significantly improved with the help of their excellent suggestions. We have submitted a revised version. Further we have added a track-changes docx document showing all changes that have been made during the revision process. In the following we will go into the individual points of the reviewers.

Response to reviewer comments:

Reviewer #1 (Remarks to the Author):

1. Line 193: The same effect---- is inappropriate, as IsPETase and LCC have Y at this position. Probably you would like to mean that mutation of Y to F and A in both enzymes have the same effect of F to Y mutation of PHL7.

Response 1-1: We changed the corresponding sentence to: "Substitution at this position with alanine also led to reduced activity in IsPETase and LCC "

2. Numbers of supplementary figures should be revised: Two figure-1s exist and the 2nd Figure 1 to Figure 3 should be Figures 2 to 4.

Response 1-2: We have checked that all numbers are correct in the current manuscript.

3. Line 234: Should be "Exchange of M161 or I208 in IsPETase--.

Response 1-3: We changed the corresponding sentence to: "Exchange of M161 or I208 (M132 and I179 in PHL7) to alanine reduced the BHET-hydrolytic activity of in IsPETase to 52 % and 46 %, respectively."

4. Lines 297-: In references 47 and Kawai et al. (AMB Express, 12: 134 (2022))), it is described that replacement of F with A and I remarkably decreased the expression of mutant enzymes, indicating that F is crucial to Cut190. This might be due to the different amino acids interacting with F in Cut190, thereby leading to different binding energy of F in PHL7 and Cut190.

Response 1-4: This is true, a similar effect has not been reported for other variants with similar mutations. However, this can have several causes, e.g. higher toxicity for the host cell. We decided to exclude the expression-effect by Kawai et al..

5. Conclusions: Supplementary Fig. 7 does not include double-gain mutants. Why not them?

Response 1-5: We have focused on a structural analysis and comparison between different types of polyester-degrading enzymes to better understand their sequence/function relationship. This work should be the fundamentals for future optimizing work.

Reviewer #2 (Remarks to the Author):

The points raised previously were answered satisfactorily.

Minor points remaining are:

(1) Legend to Fig. S1: PDB no. for PHL7-TPA crystal structure is missing; probably not assigned yet?

Response 2-1: The accession numbers have been added and will be made accessible automatically upon publication.

(2) I find former Fig. S6 much easier to understand than Figs. 3a and d. If these figures are redundant, why are they presented in the main text and, additionally, as supplementary figure?

Response 2-2: we choose to show Fig. 3a and d in the main text as these figures contain more information (Fig. 3a contains rates after 4 h and 8 h, also the standard deviation and distribution of samples is shown. However, we would like to give this decision to the editors.

(3) It was a bit irritating that the authors mentioned in their rebuttal letter page and line numbers of the original manuscript rather than of the revised version which contains the additions/changes that they referred to.

Response 2-3: We apologize for this inconvenience.